# A naturally occurring mini-alanyl-tRNA synthetase

Titi Rindi Antika[1,4], Dea Jolie Chrestella[1,4], Yi-Kuan Tseng[2], Yi-Hung Yeh[3], Chwan-Deng Hsiao[3] & Chien-Chia Wang [1✉]

Alanyl-tRNA synthetase (AlaRS) retains a conserved prototype structure throughout its biology, consisting of catalytic, tRNA-recognition, editing, and C-Ala domains. The catalytic and tRNA-recognition domains catalyze aminoacylation, the editing domain hydrolyzes mischarged tRNA$^{Ala}$, and C-Ala——the major tRNA-binding module——targets the elbow of the L-shaped tRNA$^{Ala}$. Interestingly, a mini-AlaRS lacking the editing and C-Ala domains is recovered from the Tupanvirus of the amoeba *Acanthamoeba castellanii*. Here we show that Tupanvirus AlaRS (TuAlaRS) is phylogenetically related to its host's AlaRS. Despite lacking the conserved amino acid residues responsible for recognition of the identity element of tRNA$^{Ala}$ (G3:U70), TuAlaRS still specifically recognized G3:U70-containing tRNA$^{Ala}$. In addition, despite lacking C-Ala, TuAlaRS robustly binds and charges micro$^{Ala}$ (an RNA substrate corresponding to the acceptor stem of tRNA$^{Ala}$) as well as tRNA$^{Ala}$, indicating that TuAlaRS exclusively targets the acceptor stem. Moreover, this mini-AlaRS could functionally substitute for yeast AlaRS in vivo. This study suggests that TuAlaRS has developed a new tRNA-binding mode to compensate for the loss of C-Ala.

[1] Department of Life Sciences, National Central University, Zhongli District, Taoyuan 320317, Taiwan. [2] Graduate Institute of Statistics, National Central University, Zhongli District, Taoyuan 320317, Taiwan. [3] Institute of Molecular Biology, Academia Sinica, Nankang District, Taipei 11529, Taiwan. [4]These authors contributed equally: Titi Rindi Antika, Dea Jolie Chrestella. ✉email: dukewang@cc.ncu.edu.tw

An aminoacyl-tRNA synthetase (aaRS) attaches a specific amino acid to one of its cognate tRNAs to form an aminoacyl-tRNA. This charged tRNA is then delivered to the ribosome for translation of the genetic code[1]. Therefore, aminoacyl-tRNA plays a key role in DNA translation——the expression of genes to make proteins. Many eukaryotic aaRSs recruit N- or C-terminal appended domains to expand their nontranslational functions during evolution. Hence, even aaRSs of the same amino acid specificity often possess distinct structural organizations between prokaryotes and eukaryotes. However, AlaRS has never recruited a new domain during evolution[2]. Its prototype structure consists of the catalytic, tRNA-recognition, editing, and C-terminal (C-Ala) domains[3]. AlaRS exists as a homodimer (or tetramer) in prokaryotes[4,5] and a monomer in eukaryotes[6]. For instance, the hyperthermophilic archaeon *Archaeoglobus fulgidus* AlaRS (AfAlaRS) is a homodimer of 906 amino acid residues and genetically divided into two parts. Residues 1-739 include the catalytic, tRNA-recognition, and editing domains, while residues 736-906 comprise the C-Ala domain[7]. The N-terminal two domains are often referred to as the aminoacylation domain, which is responsible for alanylation of tRNA$^{Ala}$, and the editing domain is responsible for removal of the mischarged amino acids from tRNA$^{Ala}$. Defects in editing activity leads to misacylation, mistranslation, and cytotoxicity in yeast[8] and neural degeneration in mice[9]. In contrast to the other three domains, C-Ala highly diverges among species. Prokaryotic C-Ala consists of a helical and a globular subdomain[4] and constitutes the major tRNA-binding module[10]. The helical subdomain mediates dimer formation, whereas the globular subdomain targets the elbow of the L-shaped tRNA$^{Ala}$[11]. Despite eukaryotic C-Ala possessing a similar architecture, its helical subdomain consists of three α-helices instead of two. Human C-Ala binds DNA robustly but not tRNA$^{Ala}$, whereas *C. elegans* C-Ala binds both DNA and tRNA$^{Ala}$ robustly[6,11,12].

Identity elements of tRNA may be single nucleotides, base pairs, or post-transcriptional modifications, and often reside in the acceptor stem and anticodon loop. AlaRS establishes tRNA$^{Ala}$ identity via a unique G3:U70 wobble base pair in the acceptor stem. G3:U70 exists in all tRNAs$^{Ala}$, ranging from *E. coli* to human cytoplasm, but is missing from all non-tRNAs$^{Ala}$[13–15]. Recognition of this identity element is mediated by two conserved amino acid residues, Asn (N) and Asp (D) (as N359 and D450 in *A. fulgidus* and as N303 and D400 in *E. coli*), in the tRNA-recognition domain. These two amino acid residues form specific hydrogen bonds with the G:U base pair, thereby establishing its identity as the alanine acceptor[11]. The amide nitrogen of the Asn side chain contacts O4 of U70 (from major-grove side), while the carboxyl side chain and the backbone carbonyl of Asp make hydrogen bonds with the 2-amino group of G3 (from minor-grove side)[15,16].

Tupanvirus belongs to the family of giant viruses *Mimiviridae*. Its genome contains 1400–1500 kb of double-strand linear DNA[17]. Based on the size and sequence of its genome, Tupanvirus can be divided into two strains: soda lake and deep ocean. Both strains were first identified to replicate in *Acanthamoeba castellanii* and *Vermamoeba vermiformis*[18], and later in *Acanthamoeba polyphaga*, *Acanthamoeba griffin*, *Dyctistelium discoideum*, and *Willartia magna*[19]. Interestingly, Tupanvirus possesses a complete set of translation-related genes, including 20 aaRSs and tRNA isoacceptors associated with all amino acids, tRNA and mRNA maturation enzymes, translation initiation, elongation, and releasing factors. Paradoxically, however, Tupanvirus does not share the same codon preference with its natural host *A. castellanii*. Instead, the Tupanvirus genome's codons match its own tRNA isoacceptors[18], suggesting that Tupanvirus tRNA isoacceptors may be responsible for translation of its own genome.

Among the 20 Tupanvirus aaRSs, AlaRS was of particular interest, as it contains only the aminoacylation domain. Moreover, it lacks the conserved N and D residues in its tRNA-recognition domain. This piqued our curiosity as to whether and how this mini-AlaRS can function normally. Our results showed that Tupanvirus AlaRS (TuAlaRS) can bind and charge micro$^{Ala}$ (an RNA substrate corresponding to the acceptor stem of tRNA$^{Ala}$) as well as tRNA$^{Ala}$ specifically and robustly, an unexpected feature for a C-Ala-defective AlaRS. Conceivably, this mini-AlaRS has evolved a new tRNA-binding mode to counterbalance the loss of its C-Ala.

## Results

**TuAlaRS harbors only the aminoacylation domain.** Unlike its *A. fulgidus* homologue, TuAlaRS possesses only the catalytic and tRNA-recognition domains (Fig. 1a). Motifs 1, 2, and 3 are still conserved in its catalytic domain. However, the conserved G3:U70-recognition amino acid residues N and D are missing from its tRNA-recognition domain. Instead, P321 and T416 are located at the corresponding positions in sequence alignment (Fig. 1b). To get a deeper insight, a structural model of TuAlaRS (with only 461 amino acid residues) was built using AlphaFold2 [20] and compared to the aminoacylation domain of *A. fulgidus* AlaRS (AfAlaRS-N484) obtained from AfAlaRS/tRNA$^{Ala}$ complex (PDB ID 3WQY). Both structural models were visualized and further modified using EzMol[21]. TuAlaRS folds into a three-dimensional structure closely resembling that of AfAlaRS-N484 (Fig. 1c). P321 and T416 of TuAlaRS are located at almost the same positions as the conserved N and D residues of AfAlaRS-N484. Nevertheless, there are still some notable differences between these two structures, in particular at their C-termini. The C-terminal region of AfAlaRS-N484 (aa 456–484) folds into two short α-helices, while the corresponding region of TuAlaRS (aa 432–461) folds into a long α-helix.

Four Tupanvirus tRNA$^{Ala}$ isoacceptors have been recovered, each of which carries a G3:U70 base pair (Supplementary Fig. 1). One of them has a D-loop larger than normal, while the rest of them carry a normal-sized D-loop. The sequences in the TψC-loop are conserved, while those in the D-loop are relatively diverged. Nevertheless, the bases that are important for the elbow formation (such as G$^{18}$G$^{19}$ in the D-loop and ψ$^{55}$C$^{56}$ in the TψC-loop) are still conserved, suggesting that they are likely to fold into the L-shaped structure[22].

**TuAlaRS charges tRNA$^{Ala}$ and micro$^{Ala}$ efficiently and specifically.** As TuAlaRS lacks the major tRNA-binding module (C-Ala), we wondered whether this enzyme can charge tRNA$^{Ala}$ efficiently. To this end, we purified the His$_6$-tagged TuAlaRS protein from an *E. coli* transformant and carried out an aminoacylation assay using in vitro-transcribed *E. coli* tRNA$^{Ala}$ (EctRNA$^{Ala}$) as the substrate. To our surprise, this mini-AlaRS was almost as efficient as EcAlaRS in charging EctRNA$^{Ala}$ (with ~2-fold difference in charging activity) (Fig. 2a, b). In addition, despite lacking the conserved N and D residues in its tRNA-recognition domain, TuAlaRS was still specific to G3:U70 and failed to charge EctRNA$^{Ala}$ with G3:C70 or A3:U70 to a detectable level (Fig. 2a).

We next tested whether TuAlaRS can efficiently charge a tRNA$^{Ala}$ mimic without the elbow. Pursuant to this objective, an *E. coli* microhelix$^{Ala}$ (Ec-micro$^{Ala}$, which corresponds to the acceptor stem of EctRNA$^{Ala}$) was used as the substrate (Supplementary Fig. 1, TuAlaRS charged this microhelix to a higher level than did EcAlaRS when the same concentration of enzymes was used (Fig. 2c, d). As a matter of fact, TuAlaRS charged EctRNA$^{Ala}$ and Ec-micro$^{Ala}$ to a similar level (Fig. 2a, c). Not surprisingly, mutation of G3:U70 to G3:C70 or A3:U70 in

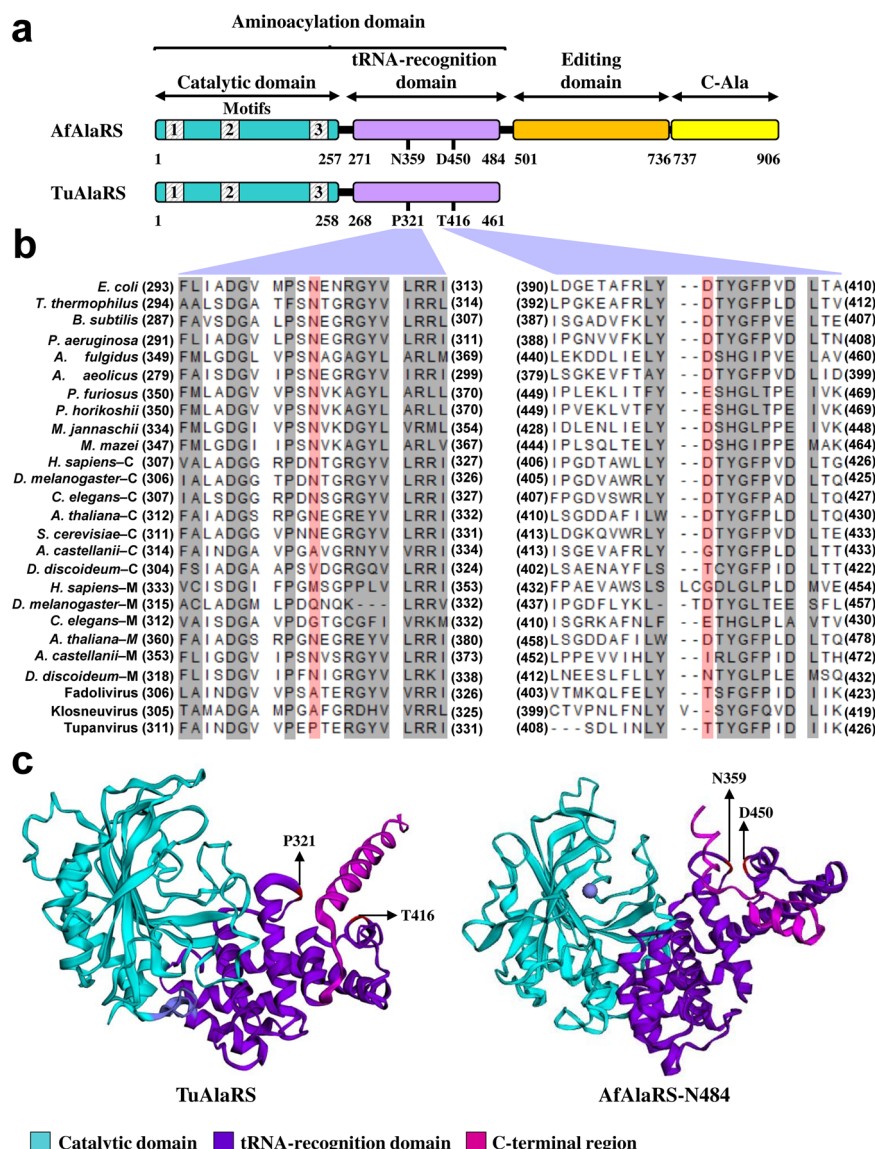

**Fig. 1 Domain organization of TuAlaRS. a** Schematic representation of TuAlaRS and AfAlaRS. The relative positions of the four domains are marked. **b** Alignment of the sequences flanking the G3:U70-recognition amino acid residues N and D (highlighted in red) in the tRNA-recognition domain. **c** Structural modeling of TuAlaRS in comparison with AfAlaRS-N484. The catalytic and tRNA-recognition domains were colored in light-blue and purple, respectively. The C-terminal helix of the tRNA-recognition domain was highlighted in pink.

Ec-micro$^{Ala}$ dramatically impaired its aminoacylation by TuA-laRS (Fig. 2c). To determine whether Tupanvirus tRNA$^{Ala}$ (TutRNA$^{Ala}$) isoacceptors are good substrates for TuAlaRS and EcAlaRS, we arbitrarily chose one of the TutRNA$^{Ala}$ isoacceptors (second from the left in Supplementary Fig. 1A) as the substrate for aminoacylation. As shown in Supplementary Fig. 2, both enzymes could charge TutRNA$^{Ala}$, albeit to a level slightly lower than that of EctRNA$^{Ala}$ (2–4-fold). In addition, we also checked whether TuAlaRS can efficiently charge its host (*A. castellanii*) tRNA$_n^{Ala}$ (Supplementary Fig. 1). The results showed that TuAlaRS charges *A. castellanii* tRNA$_n^{Ala}$ as well as EctRNA$^{Ala}$ (Supplementary Fig. 3).

**TuAlaRS binds both tRNA$^{Ala}$ and micro$^{Ala}$ robustly.** TuAlaRS's ability to charge both EctRNA$^{Ala}$ and Ec-micro$^{Ala}$ efficiently implies that this mini-AlaRS can bind both ligands robustly. To take a closer look, we carried out an electrophoretic mobility shift assay (EMSA) using EctRNA$^{Ala}$ and Ec-micro$^{Ala}$ as the ligands. As shown in Fig. 3a, TuAlaRS bound EctRNA$^{Ala}$ robustly (with a

$K_d$ value of 0.3 μM). Mutation of G3:U70 to G3:C70 or A3:U70 reduced its binding affinity 2–3-fold (Fig. 3a). EcAlaRS could also bind EctRNA$^{Ala}$ robustly (with a $K_d$ value of 1.0 μM; Fig. 3b). However, unlike the scenario of TuAlaRS, mutation at G3:U70 had little effect on its binding by EcAlaRS (~1.3-fold reduction in binding affinity) (Fig. 3b). Thus, mutation at G3:U70 had a stronger impact on binding by TuAlaRS than by EcAlaRS. A similar scenario was observed in mutation of the 3′-CCA end, in which deletion of the 3′-CCA end or substituting the 3′-CCA end with UUG reduced its binding by TuAlaRS 3–4-fold (Fig. 3a), but hardly affected its binding by EcAlaRS (Fig. 3b). The fact that deletion and substitution of the 3′-CCA end showed a comparable effect on TuAlaRS's binding implies that the 3′-CCA end is important for recognition by TuAlaRS.

We next checked whether TuAlaRS can bind Ec-micro$^{Ala}$ robustly. Consistent with an earlier report[23], EcAlaRS bound Ec-micro$^{Ala}$ poorly (with a $K_d$ value of >32 μM) (Fig. 3d). However, it was interesting to find that TuAlaRS binds Ec-micro$^{Ala}$ robustly (with a $K_d$ value of 0.3 μM) (Fig. 3c). As a result, TuAlaRS bound

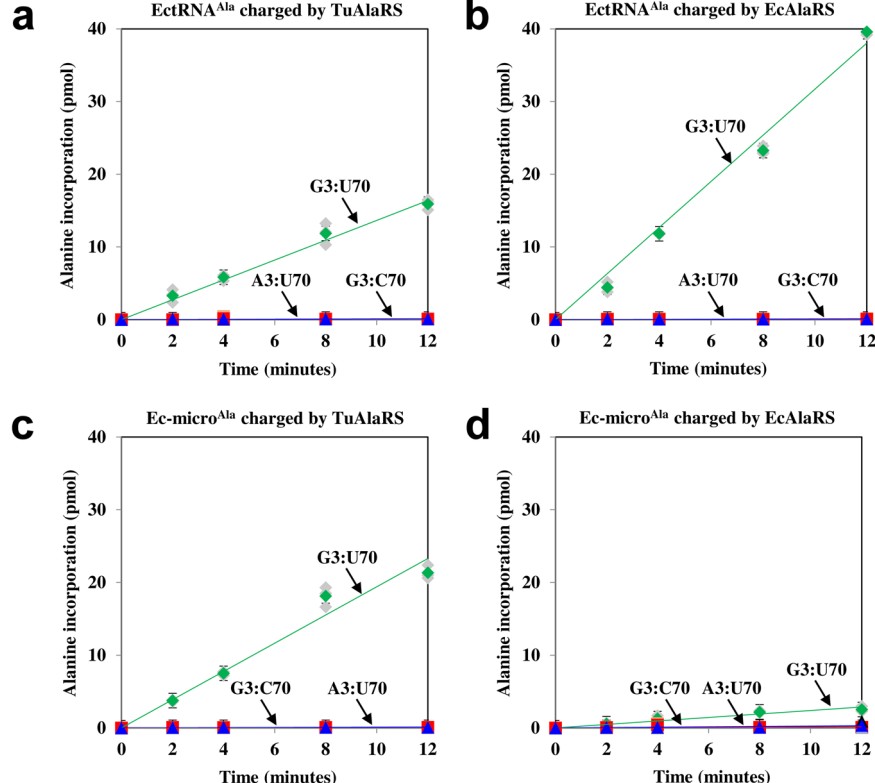

**Fig. 2 Aminoacylation assay.** The aminoacylation activities of (**a**) TuAlaRS and (**b**) EcAlaRS toward WT and mutant EctRNA$^{Ala}$ transcripts were determined by testing their ability to incorporate $^3$H-alanine into the transcripts. The aminoacylation activities of (**c**) TuAlaRS and (**d**) EcAlaRS toward WT and mutant Ec-micro$^{Ala}$ transcripts were determined under similar conditions. The enzyme concentration used in the reactions was 100 nM. Error bars are standard deviation from triplicates.

to EctRNA$^{Ala}$ and Ec-micro$^{Ala}$ equally well (Fig. 3a, c), indicating that this mini-AlaRS mainly targets the acceptor stem instead of the elbow. Mutation of G3:U70 to G3:C70 or A3:U70 reduced its binding affinity distinctly, but not drastically (~3-fold increase in $K_d$; Fig. 3c). In contrast, an EcAlaRS fragment (N461) possessing the N-terminal 461 amino acid residues bound EctRNA$^{Ala}$ and Ec-micro$^{Ala}$ poorly (with $K_d$ values of >32 μM) (Supplementary Fig. 4), emphasizing the indispensable role of C-Ala in tRNA$^{Ala}$ binding by EcAlaRS. Hence, the major binding affinity of EcAlaRS for tRNA$^{Ala}$ arises from C-Ala targeting the elbow[10], whereas the major binding affinity of TuAlaRS for tRNA$^{Ala}$ arises from the mini-synthetase targeting the acceptor stem (Fig. 3). This might also explain why mutations in the acceptor stem have a stronger effect on binding by TuAaRS.

**TuAlaRS fails to charge *C. elegans* mitochondrial micro$^{Ala}$.** As reported earlier, an extra G:U base pair, G1:U72, in the acceptor stem of *C. elegans* mitochondrial tRNA$^{Ala}$ blocks its aminoacylation by *E. coli*, *S. cerevisiae*, and *C. elegans* cytoplasmic AlaRSs efficiently[12,24]. We wondered whether this extra G:U base pair has a similar effect on TuAlaRS. To this end, we carried out an aminoacylation assay using *C. elegans* mitochondrial micro$^{Ala}$ (Ce-micro$^{Ala}$) (Supplementary Fig. 1) as the substrate. As shown in Supplementary Fig. 5A, TuAlaRS charged Ec-micro$^{Ala}$ efficiently but not Ce-micro$^{Ala}$, suggesting that G1:U72 blocks TuAlaRS's aminoacylation of Ce-micro$^{Ala}$. We next checked whether this anti-determinant blocks its binding by the mini-AlaRS. Contrary to our anticipation, TuAlaRS bound Ce-micro$^{Ala}$ with affinity similar to that for Ec-micro$^{Ala}$ (with $K_d$ values of 0.3–0.4 μM) (Supplementary Fig. 5B), suggesting that this anti-determinant affects catalysis instead of binding. Perhaps G1:U72

interferes with the positioning of the 3′-CCA end of the acceptor stem in the active-site pocket.

**Kinetic parameters for aminoacylation of tRNA$^{Ala}$ and micro$^{Ala}$ by TuAlaRS.** To obtain the kinetic parameters ($K_M$ and $k_{cat}$) for TuAlaRS's aminoacylation of EctRNA$^{Ala}$ and Ec-micro$^{Ala}$, we performed a kinetic assay using EcAlaRS as a control. As shown in Table 1, TuAlaRS charged EctRNA$^{Ala}$ and Ec-micro$^{Ala}$ with similar efficiency (<2-fold difference in $k_{cat}/K_M$), whereas EcAlaRS highly preferred EctRNA$^{Ala}$ over Ec-micro$^{Ala}$ (~68-fold difference in $k_{cat}/K_M$). In addition, EcAlaRS and TuAlaRS charged EctRNA$^{Ala}$ with similar efficiency (~2-fold difference in $k_{cat}/K_M$), but TuAlaRS charged Ec-micro$^{Ala}$ with a much higher efficiency than did EcAlaRS (~26-fold difference in $k_{cat}/K_M$). Moreover, mutation of G3:U70 to G3:C70 or A3:U70 in EctRNA$^{Ala}$ or Ec-micro$^{Ala}$ drastically reduced its aminoacylation efficiency (~100-fold reduction in $k_{cat}/K_M$) by TuAlaRS, with a major impact on $k_{cat}$, a scenario reminiscent of the EcAlaRS-catalyzed tRNA$^{Ala}$ aminoacylation[15]. Thus, discrimination of G3:U70 by TuAlaRS mainly occurs on the $k_{cat}$ level rather than on the $K_M$ level.

**P321 and T416 are dispensable for aminoacylation.** As shown in Fig. 1, the highly conserved N and D are absent from the tRNA-recognition domain of TuAlaRS. Instead, P321 and T416 are located at the corresponding positions in sequence alignment (Fig. 1b). In addition, structural modeling showed that the P and T residues are located at almost the same positions as the N and D residues of AfAlaRS-N484 (Fig. 1c). To check whether these two amino acid residues still play a similar role in recognition of G3:U70, we mutated P and T to A individually and assayed their aminoacylation activities. As expected, mutation at P or T had

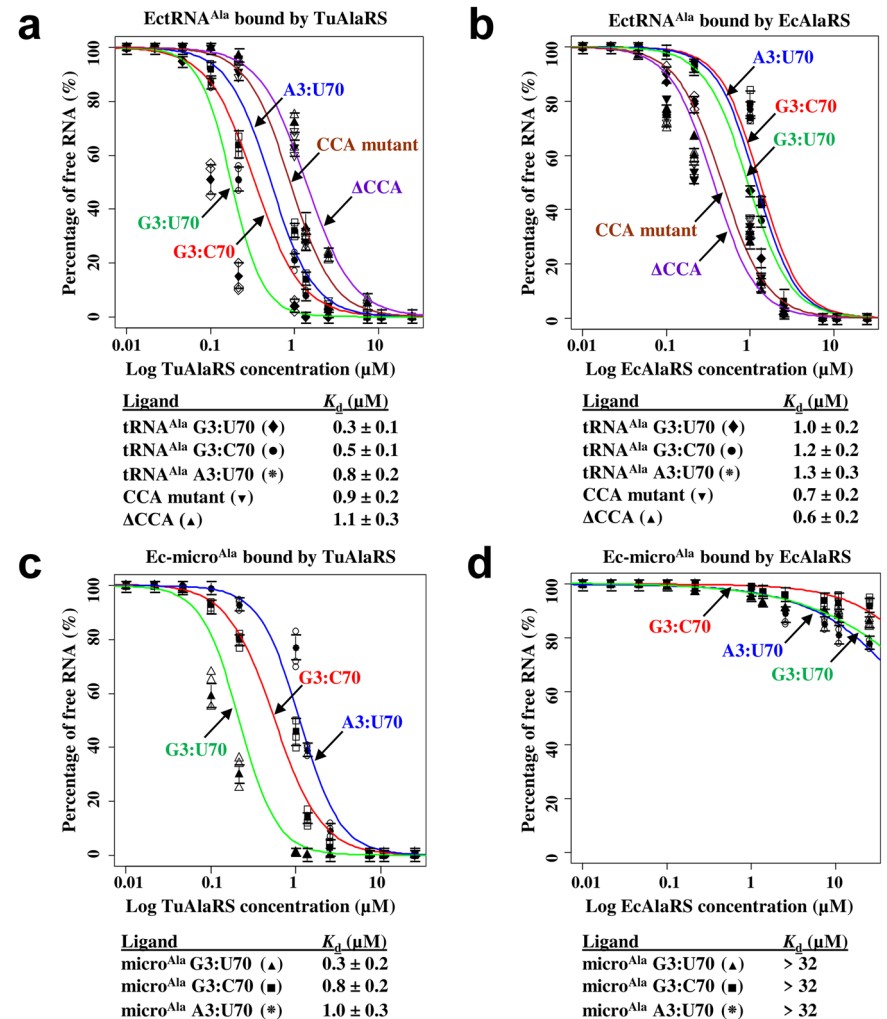

**Fig. 3 Binding of TuAlaRS and EcAlaRS to EctRNA$^{Ala}$ and Ec-micro$^{Ala}$ transcripts.** The binding affinities of (**a**) TuAlaRS and (**b**) EcAlaRS for WT and mutant EctRNA$^{Ala}$ transcripts were determined by an EMSA with protein concentrations ranging from 32 to 0.0625 μM. The binding affinities of (**c**) TuAlaRS and (**d**) EcAlaRS for WT and mutant Ec-micro$^{Ala}$ transcripts were determined under similar conditions. Error bars are standard deviation from triplicates.

**Table 1 Kinetic parameters for aminoacylation of EctRNA$^{Ala}$ and Ec-micro$^{Ala}$ by EcAlaRS and TuAlaRS.**

| Enzyme | EctRNA$^{Ala}$ | $K_M$ (μM) | $k_{cat}$ ($\times 10^{-3}$ s$^{-1}$) | $k_{cat}/K_M$ ($\times 10^{-3}$ μM$^{-1}$s$^{-1}$) |
|---|---|---|---|---|
| EcAlaRS | G3:U70 | 1.3 ± 0.3 | 70 ± 6.1 | 54 |
| TuAlaRS | G3:U70 | 1.6 ± 0.3 | 41 ± 4.3 | 26 |
|  | G3:C70 | 4.7 ± 0.6 | 2 ± 0.4 | 0.4 |
|  | A3:U70 | 3.2 ± 0.5 | 1 ± 0.3 | 0.3 |

| Enzyme | Ec-micro$^{Ala}$ | $K_M$ (μM) | $k_{cat}$ ($\times 10^{-3}$ s$^{-1}$) | $k_{cat}/K_M$ ($\times 10^{-3}$ μM$^{-1}$s$^{-1}$) |
|---|---|---|---|---|
| EcAlaRS | G3:U70 | 154.6 ± 15.3 | 124 ± 15.5 | 0.8 |
| TuAlaRS | G3:U70 | 2.3 ± 0.3 | 49 ± 5.2 | 21 |
|  | G3:C70 | 1.8 ± 0.3 | 0.4 ± 0.1 | 0.2 |
|  | A3:U70 | 4.1 ± 0.6 | 1 ± 0.2 | 0.2 |

little effect on the enzyme's tRNA charging activity and specificity. Similar to the WT enzyme, P321A and T416A could efficiently charge G3:U70-containing Ec-micro$^{Ala}$, but not G3:C70 or A3:U70-containing Ec-micro$^{Ala}$ (Supplementary Fig. 6). Thus, P321 and T416 appear to be dispensable for G3:U70 recognition.

**TuAlaRS can functionally rescue a yeast *ALA1* knockout (KO) strain**. To investigate whether TuAlaRS can function in vivo as an AlaRS enzyme, we cloned the gene encoding TuAlaRS into a yeast shuttle vector (with a *TEF1* promoter) and transformed the resultant construct into a yeast *ALA1* (which encodes AlaRS) KO strain to test whether this mini-AlaRS can substitute for yeast AlaRS and rescue the growth defect of the KO strain. As shown in Fig. 4a, the KO strain could not grow in the absence of a functional AlaRS on 5-FOA. Remarkably, TuAlaRS was able to restore the growth phenotype of the KO strain on 5-FOA, albeit not as efficiently as yeast AlaRS. This result suggests that TuAlaRS can charge yeast tRNA$_n$$^{Ala}$ to a level sufficient for growth and that a lack of the editing domain in this mini-AlaRS does not cause cytotoxicity. Western blotting further showed that TuAlaRS is expressed to a level distinctly lower than that of yeast AlaRS under the conditions used (Fig. 4b). Thus, TuAlaRS can be a functional enzyme in vivo. In contrast, while EcAlaRS effectively rescued the growth defect of the KO strain, N461, an EcAlaRS fragment comparable to TuAlaRS, failed to do so. We wondered whether TuAlaRS shows toxicity under conditions with extra serine or glycine. To this end, we carried out complementation assays on 5-FOA with additional serine. The results showed that addition of extra serine (up to 5 mM) does not cause any toxic

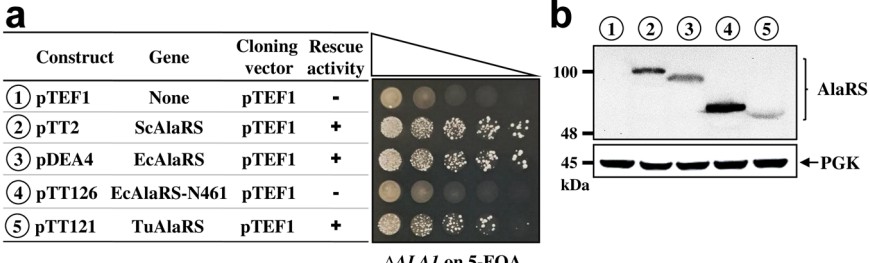

**Fig. 4 Rescue of a yeast *ALA1* KO strain by TuAlaRS. a** Complementation assay. TuAlaRS's rescue activity was determined by transforming the test plasmid into a yeast *ALA1* KO strain and plating the resultant transformants on 5-FOA. The symbols "+" and "−" denote positive and negative complementation, respectively. **b** Western blotting. The protein expression levels of these constructs were determined by Western blotting using an anti-His$_6$-tagged antibody as the probe. Constructs used in (**a**, **b**) are numbered for clarity.

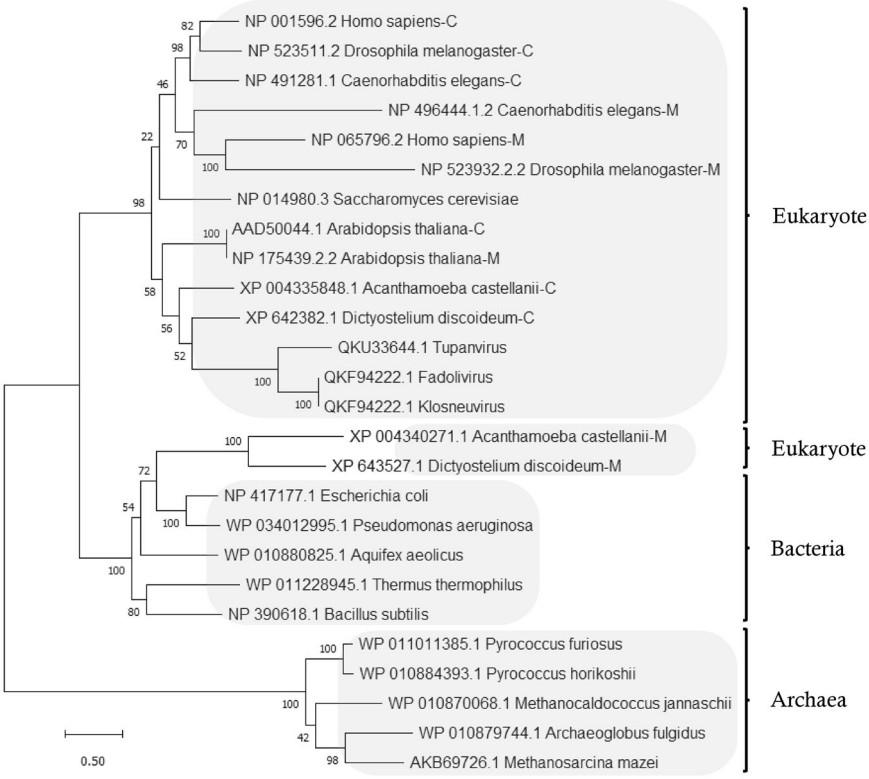

**Fig. 5 Phylogenetic analysis of AlaRSs.** The evolutionary history was inferred by using the Maximum Likelihood method and Le_Gascuel_2008 model. Initial tree(s) for the heuristic search were obtained automatically by applying Neighbor-Join and BioNJ algorithms to a matrix of pairwise distances estimated using the JTT model. A discrete Gamma distribution was used to model evolutionary rate differences among sites. This analysis involved 26 amino acid sequences. Evolutionary analyses were conducted in MEGA11.

phenotype in the yeast knockout strain carrying TuAlaRS (Supplementary Fig. 7), suggesting that this enzyme has tolerable or low mischarging activity.

**Phylogenetic relationship of TuAlaRS with other AlaRSs.** To trace the evolutionary origin of TuAlaRS and its relationships with other AlaRSs, we aligned the protein sequences of representative AlaRSs recovered from all three domains of life and built a phylogenetic tree[25,26]. As shown in Fig. 5, except for the mitochondrial AlaRSs of *A. castellanii* and *D. discoideum*, which were clustered into the bacterial group, all cytoplasmic and mitochondrial isoforms of eukaryotic AlaRSs tested were clustered into the same group, suggesting that these isoforms descended from common predecessors through gene duplication (Fig. 5). In addition, this main eukaryotic group showed a higher affinity for the bacterial than for the archaeal group, suggesting a bacterial origin for the eukaryotic

AlaRSs. As for TuAlaRS, it was closely associated with *A. castellanii* and *D. discoideum* cytoplasmic AlaRSs and belonged to the main eukaryotic group. As *A. castellanii* and *D. discoideum* are its natural hosts, this finding implies that TuAlaRS was horizontally transferred from its host genome and the gene encoding this enzyme was reduced afterwards.

**Discussion**

**TuAlaRS displays a G3:U70 specificity.** AlaRS is the only aaRS that retains a prototype structure throughout its biology. However, *Nanoarchaeum equitans* harbors a split AlaRS gene that encodes an α and a β subunit[27]. The α-subunit, comprising an intact catalytic domain and part of the tRNA-recognition domain, charges tRNA$^{Ala}$ with a low activity and in a non-G3:U70-specific manner. The β-subunit comprises the rest of the tRNA-recognition domain along with the editing and C-Ala domains.

Addition of the β subunit into the α subunit-catalyzed aminoacylation reaction enhances its activity drastically and confers a G3:U70 specificity. Unlike *N. equitans* AlaRS, TuAlaRS contains an intact catalytic domain and an intact tRNA-recognition domain but lacks the editing and C-Ala domains completely (Fig. 1a). This mini-AlaRS charges G3:U70-containing tRNA$^{Ala}$ and micro$^{Ala}$ efficiently and specifically (Fig. 2 and Table 1).

Paradoxically, TuAlaRS lacks the conserved N and D residues in its tRNA-recognition domain; instead P and T residues are located at the corresponding positions (Fig. 1). However, mutation at either amino acid residue had little effect on the enzyme's tRNA charging activity or specificity (Supplementary Fig. 6). Clearly, TuAlaRS uses a recognition mode distinct from *E. coli* (which selects G3:U70 through a direct positive recognition) or eukarya/archaea (which repels a non-G3:U70 base pair) AlaRS[15]. Perhaps, these two amino acids play a role in the enzyme's structural stability or flexibility. To facilitate the acceptor stem binding, the enzyme might select the universal identity element through a new site/mode. It is worth mentioning that *C. elegans* mitochondrial AlaRS also lacks the conserved N/D, but the enzyme still exhibits a G3:U70 specificity[12,24]. Human mitochondrial AlaRS, on the other hand, presents a completely different scenario, in which the conserved N/D residues are replaced by M/G and its cognate tRNA lacks G3:U70 (Fig. 1). As it turns out, this enzyme discriminates human mitochondrial tRNA$^{Ala}$ mainly through recognition of its variable loop[28]. In addition, *Drosophila melanogaster* mitochondrial AlaRS recognizes a shifted G:U base pair —— G2:U71[29].

In the case of tyrosyl-tRNA synthetase (TyrRS), an insertion peptide that splits the active site-containing domain determines the species-specific acylation. Substrate specificity can be switched by transplanting part of the insertion from human TyrRS into *E. coli* TyrRS and, reciprocally, by transplanting the counterpart from the *E. coli* enzyme into the human enzyme[30]. Unfortunately, there is no insertion peptide found in the active site-containing domain of AlaRS. Instead, an anti-determinant, G1:U72, is found in the acceptor stem of the *C. elegans* mitochondrial tRNA$^{Ala}$. *E. coli* and Tupanvirus AlaRSs fail to charge this tRNA because of the presence of G1:U72. Charging is effectively blocked by the 4-ketooxygen of U72 in the major grove[31]. However, adaptations make *C. elegans* mitochondrial AlaRS insensitive to the presence of the 4-ketooxygen. Extensive alanine-scanning mutagenesis of motif 2 of EcAlaRS provided little evidence for important contacts between tRNA$^{Ala}$ and this segment of the protein[32], which is largely consistent with the observation made in the cocrystal structure of *A. fulgidus* AlaRS:tRNA$^{Ala}$[16]. Perhaps, it is the flexibility of the motif 2 loop, instead of the amino acid residues therein, that enables the *C. elegans* mitochondrial enzyme to accommodate the 4-ketooxygen.

**Despite lacking C-Ala, TuAlaRS binds tRNA$^{Ala}$ robustly**. In *E. coli*, C-Ala constitutes the major tRNA-binding module, targeting the elbow of the L-shaped tRNA$^{Ala}$[10]. As a result, EcAlaRS charges EctRNA$^{Ala}$ efficiently but not Ec-micro$^{Ala}$ (by a ~60-fold difference)[23] (also repeated and shown in Table 1). Deletion of C-Ala from EcAlaRS reduces its binding affinity for tRNA$^{Ala}$ drastically (up to 13-fold)[10]. A similar scenario was observed for *C. elegans* cytoplasmic AlaRS[12]. Despite the fact that human cytoplasmic AlaRS can charge tRNA$^{Ala}$ efficiently in the absence of its C-Ala[6], it is unclear whether this enzyme can also efficiently charge micro$^{Ala}$. In addition, a truncated form of *Aquifex aeolicus* AlaRS that contains only the aminoacylation domain (N453) can efficiently charge a G3:U70-containing minihelix (an RNA substrate corresponding to the acceptor stem plus the TψC arm)[33], suggesting that its C-Ala is dispensable for tRNA binding. We

demonstrated herein that TuAlaRS can bind and charge micro$^{Ala}$ and tRNA$^{Ala}$ with a similarly high efficiency (Figs. 2 and 3 and Table 1). Clearly, TuAlaRS exclusively targets the acceptor stem of tRNA$^{Ala}$. However, it should be emphasized that G3:U70 in the acceptor stem still represents the major identity element for TuAlaRS's recognition of tRNA$^{Ala}$ (Fig. 2). Mutation at this base pair primarily affects $k_{cat}$ (catalysis) instead of $K_M$ (apparent affinity) for aminoacylation (Table 1), a scenario reminiscent of EcAlaRS[34]. To the best of our knowledge, TuAlaRS is the only naturally occurring aaRS that contains only the aminoacylation domain and can function normally.

According to the *A. fulgidus* AlaRS:tRNA$^{Ala}$ complex, the amino acid residues $^{193}$GGG$^{195}$ act as the route separator between reactive and non-reactive CCA-ends[16]. Mutation of any of these three Gly residues causes mis-aminoacylation of G3:C70-containing tRNA$^{Ala}$[34]. Interestingly, the second and third Gly residues are conserved in TuAlaRS and other AlaRSs, implying that the remaining Gly residues play a similar role as the route separator. Moreover, unlike *E. coli* AlaRS, which binds the CCA-containing and CCA-deletion (or substitution) tRNAs$^{Ala}$ with a similar affinity, TuAlaRS distinctly prefers the CCA-containing tRNA$^{Ala}$ (Fig. 3a, b), lending further support to the hypothesis that TuAlaRS has gained additional specific contacts with the acceptor stem through the CCA-end.

**TuAlaRS may have lost its editing and C-Ala domains later during evolution**. Our phylogenetic analysis suggested that TuAlaRS is closely related to its hosts' cytoplasmic AlaRSs (Fig. 5), consistent with the hypothesis that Tupanvirus acquired most of its genes through horizontal transfer from its host genomes[18,35]. However, gene acquisition in giant viruses such as Tupanvirus is often accompanied by gene loss or reduction in response to environmental changes during evolution. Thus, TuAlaRS's smaller size may have resulted from gene reduction during evolution. Examples of gene reduction were also found in TyrRS of *Acanthamoeba polyphaga* mimivirus and many proteins of Chlorella virus[36,37]. Regardless of the detailed interpretation, our study suggests that TuAlaRS has developed a new binding mode to counterbalance the loss of C-Ala. Instead of targeting the elbow, this mini-AlaRS targets the acceptor stem (Fig. 3). In a sense, TuAlaRS exemplifies an ancient-like aaRS, which possesses only the aminoacylation domain and identifies its cognate tRNA through recognition of an operational RNA code (a specific sequence or structure) in the acceptor stem[38].

## Methods

**Construction of plasmids**. For expression in *E. coli*, the gene encoding TuAlaRS (GenBank: MF405918.2; protein ID: QKU33644.1) was codon optimized for *E. coli* and synthesized by Omics Biotechnology (Taiwan). The synthesized gene was cloned into the NdeI/XhoI sites of pET21b (an *E. coli* expression vector) and the resultant plasmid was transformed into an *E. coli* strain, BL21-CodonPlus(DE3). The resultant transformants were induced with 1 mM isopropyl β-D-1-thiogalactopyranoside for 4 h at 30 °C. The His$_6$-tagged protein was purified to homogeneity through Ni-NTA affinity chromatography followed the protocols provided by the manufacture of the Ni-NTA resin (Qiagen, Hilden, Germany)[39]. For expression in yeast, the gene encoding TuAlaRS was codon optimized for *S. cerevisiae* and cloned in pTEF1 (a yeast shuttle vector with a strong *TEF1* promoter).

**Electrophoretic mobility shift assay (EMSA)**. Protein-RNA binding affinities were determined by an EMSA, as described earlier[10]. The RNA substrate was $^{32}$P-radiolabeled at the 5′-end[40]. For each complex formation, ~10,000 cpm of RNA (diluted with cold RNA to 0.1 μM final concentration) was incubated for 10 min on ice with variable concentrations of AlaRS in a total volume of 20 μl containing 20 mM HEPES, pH 7.5, 20 mM KCl, 5 mM MgCl$_2$, and 2 mM DTT. Glycerol was added to a final concentration of 5% in each sample prior to loading on a native 5% 29:1 polyacrylamide gel. The gel was run at 100 V for 30–45 min at 4 °C in 0.5× TBE and then dried. The dried gel was exposed to X-ray film at −80 °C with an intensifying screen. The signal intensity was compared through ImageJ to measure the remaining free $^{32}$P-RNA.

**Complementation assay on FOA**. Construction of a yeast haploid *ALA1* KO strain (*MAT*α, *ala1::kanMX4, his3Δ1 leu2Δ0 lyS3Δ0 ura3Δ0*) was described earlier[41]. To examine whether a heterologous AlaRS gene can functionally substitute for the cytoplasmic activity of yeast *ALA1* (which encodes both the cytoplasmic and mitochondrial forms of AlaRS), a test plasmid carrying the target gene and a *LEU2* marker was transformed into the KO strain, and the resultant transformant was tested for its ability to grow on 5-FOA (1 mg/ml) plates. Because 5-FOA can be converted to a toxic compound in the presence of *URA3*, the transformant must evict the maintenance plasmid (carrying a WT *ALA1* gene and a *URA3* marker) on 5-FOA to survive. Thus, the transformant cannot grow on 5-FOA unless the test plasmid encodes a functional cytoplasmic AlaRS.

**In vitro transcription of tRNA^Ala and micro^Ala**. In vitro transcription of tRNA followed a standard protocol. We synthesized and cloned a DNA fragment containing a T7 promoter followed by a sequence encoding tRNA^Ala into the SmaI site of pUC18. We enriched the transcription DNA template by PCR amplification of the cloned fragment. We performed in vitro transcription at 37 °C for 3 h with 0.3 μM T7 RNA polymerase in a buffer containing 20 mM Tris-HCl (pH 8.0), 150 mM NaCl, 20 mM MgCl$_2$, 5 mM DTT, 1 mM spermidine, and 2 mM of each NTP. We purified the tRNA transcript using an 8 M urea-15% polyacrylamide gel. Following purification, ethanol precipitation, and vacuum drying, we dissolved the tRNA pellet in 1× TE buffer (20 mM Tris-HCl [pH 8.0] and 1 mM EDTA), refolded the tRNA by heating it to 80 °C for 3 min, and gradually cooled it to 55 °C before adding 2 mM of MgCl$_2$. We collected the refolded tRNA when the temperature reached ~25 °C and stored it at -80 °C. Preparation of micro^Ala followed a similar protocol.

**Aminoacylation assay**. We performed in vitro aminoacylation assays at 37 °C in a buffer containing 50 mM HEPES (pH 7.5), 50 mM KCl, 15 mM MgCl$_2$, 5 mM dithiothreitol, 10 mM ATP, 0.1 mg/ml bovine serum albumin, 5 μM tRNA^Ala (or micro^Ala), 20 μM alanine (including 1.34 μM ^3H-alanine; PerkinElmer, Waltham, MA, USA), and 100 nM AlaRS. At various times, we quenched the reactions by spotting 10 μl aliquots of the reaction mixture onto Whatman filter pads (Maidstone, Kent, UK). The filter pads had been presoaked in a solution containing 5% trichloroacetic acid and 2 mM alanine. We washed the filter pads three times for 15 min each in ice-cold 5% trichloroacetic acid and dried the pads with a heat lamp before liquid scintillation counting. We obtained aminoacylation data from three independent experiments and averaged them.

Kinetic parameters for aminoacylation of tRNA^Ala and micro^Ala by AlaRS were determined by fitting the data points directly to the Michaelis-Menten equation. Initial rates of aminoacylation were determined at 37 °C with tRNA^Ala concentration ranging from 1 to 10 μM, and micro^Ala concentrations ranging from 1 to 10 μM (for TuAlaRS) or 10–400 μM (for EcAlaRS). AlaRS concentrations ranged from 200 to 4000 nM. Active protein concentrations were determined by active site titration in the reaction cocktail containing [γ-^32P]ATP, 10 mM MgCl$_2$, 144 mM Tris-Cl (pH 7.8), 1 mM cognate amino acid, and yeast inorganic pyrophosphate (2 units/ml). The reaction was then incubated at 25 °C[42].

**Statistics and reproducibility**. The data were obtained from three independent experiments and averaged.

**Reporting summary**. Further information on research design is available in the Nature Portfolio Reporting Summary linked to this article.

## Data availability

Data supporting the findings of this study are available within the paper and its supplementary material. Uncropped/unedited gels are shown in Supplementary Fig. 8. Numerical source data are available in Supplementary Data 1.

## Code availability

There is no custom code or mathematic algorithm used in this study.

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

## Acknowledgements

This work was supported by a grant (MOST 110-2311-B-008-001) from the Ministry of Science and Technology (Taipei, Taiwan).

## Author contributions

All authors contributed to the study conception and design. Material preparation, data collection and analysis were performed by T.R.A., D.J.C., Y-K.T., Y-H.Y, C-D.H., and C-C.W. The first draft of the paper was written by T.R.A., D.J.C., and C-C.W. and all authors commented on previous versions of the paper. All authors read and approved the final paper.

## Competing interests

The authors declare no competing interests.
