## [Peer Review File · Communications Biology]

Reviewers' comments:

Reviewer #1 (Remarks to the Author):

In this paper, Antika et al. investigated the unique characters of Tupanvirus AlaRS (TuAlaRS). As a reviewer, I really enjoyed their paper. The authors presented interesting findings including: (1) Mini-AlaRS lacking the editing and C-Ala domains was recovered from the Tupanvirus of the amoeba *Acanthamoeba castellanii*. (2) Despite lacking the conserved amino acid residues responsible for recognition of the identity element of tRNAAla (G3:U70), TuAlaRS still specifically recognized G3:U70-containing tRNAAla. (3) TuAlaRS robustly bound and charged microAla (an RNA substrate corresponding to the acceptor stem of tRNAAla) as well as tRNAAla. (4) Mini-AlaRS could functionally substitute for yeast AlaRS in vivo. Such findings about TuAlaRS themselves deserve to be published in *Communications Biology*, but I would like to see the revised version of the paper for clarity according to the following points:

1. According to the complex structure of *A. fulgidus* tRNA:AlaRS, G3:U70 base pair and the hydrogen bonding contacts on the G3:U70 base pair are made with the major-groove N and minor-groove D residues. The authors should mention them more specifically in Introduction or Discussion.
2. We cannot tell that the loops (especially anticodon-loop) of Tupanvirus tRNAAla are smaller than normal (Page 4, Line 20-21 and also Fig. S1). Anticodon loop is normally composed of 7 nucleotides and "U-A" in Tupanvirus tRNAAla may be "U and A" (without base pair) as same as EctRNAAla. (I think "U and A" (without base pair) are more probable because Tupanvirus tRNAAla can be used in the ribosome.)
3. "In contrast to EcAlaRS, which charged Ec-microAla poorly (Fig. 2D), TuAlaRS charged this microhelix efficiently (Fig. 2C)" (Page 5, Line 6-7) is not appropriate. It depends on the concentration of the enzyme and the substrate used in the experiment. The author should modify the description. Because the authors also determined the kinetic parameters, the explanations should be dependent on the kinetics.
4. The fact that deletion and substitution of the 3'-CCA end showed a comparable effect on TuAlaRS's binding gave the most impact on me in their paper. The authors should make more discussion on the recognition of the 3'-CCA end by this mini-AlaRS in terms of structure. The authors can at least compare the sequence-alignments of mini-AlaRS and *A. fulgidus* AlaRS (as shown in Fig. 1B). In addition, the specific residues responsible for the CCA recognition can be obtained from the complex structure of *A. fulgidus* tRNA:AlaRS is available. Therefore, the authors should use of them for discussion.
5. *Drosophila melanogaster* mitochondrial (Dm mt) AlaRS recognizes G2:U71. In this Dm mtAlaRS, an insertion of 27 amino acids in the region of the protein that contacts the acceptor stem played an important role of G2:U71 recognition (Lovato et al., *Mol Cell*. 2004 843-851). Can the authors make a discussion about the similar insertion-based interpretation for their result, especially regarding the G1:U72 anti-determinant?
6. Regarding the G:U recognition, the conserved N and D architecture of AlaRSs can select G:U in a straightforward (bacteria) or two different unconventional (eukarya/archaea) ways (reference #13). The authors should discuss (or at least speculate) on the role of P and T in TuAlaRS, which are located at the corresponding positions as N and D, in considering above mentioned different modes for recognition of G:U pair in tRNAAlas from different species.

Minor points:

λ "GU base pair" should be written as "G:U base paper" (Page 3, Line 7; Page 6, Line 16; Page 6, Line 19; Page 9, Line 15).

Reviewer #2 (Remarks to the Author):

Antika et al. characterize Alanyl-tRNA synthetase found in a giant virus that uses amoebas as hosts, the Tupanvirus. These viruses have a complete set of aminoacyl-tRNA synthetases as well as corresponding tRNAs. By analyzing the domain structure of TuAlaRS, they authors find that it lacks both the editing domain and C-Ala, which is surprising as AlaRS is otherwise well conserved across species. Using careful biochemical characterization, they find that TuAlaRS is able to aminoacylate both its own and E. coli tRNA, but not mitochondrial tRNA, which uses a different identity element. In contrast though, TuAlaRS recognizes the stem loop along as the domain which usually recognizes the elbow region of tRNA-Ala is missing. Yeast complementation shows that this small version of AlaRS is otherwise fully functional and specific despite the reduced site.

This is a fascinating study into the minimalism of viral proteins, focusing here on the loss of two domains that were thought to be crucial for function, but can be evolutionary compensated for. I would suggest that the following points should be addressed:

1. While the authors discuss the loss of C-Ala and provide experimental evidence for how TuAlaRS evolved around this loss, the lack of the editing domain is less explored. Loss of AlaRS editing in yeast causes misincorporation of serine - is increased sensitivity towards serine toxicity observed in the TuAlaRS-complemented yeast strains? Alternatively, can increased mischarging be observed in vitro? Could host AlaRS compensate for the lack of editing by TuAlaRS or might there even be a speculated evolutionary advantage of this lack in editing?
2. Does TuAlaRS charge the tRNA-Ala isoacceptors of its host? If not, this would help support the hypothesis that the complete set of aaRSs was acquired by Tupanviruses to enable orthogonality between virus and host translation. If experimental data is difficult to obtain, a comparison between all tRNAs could be added to Figure S1.

Minor:

1. *A. fulgidus* is mentioned without context - I am assuming it is used for comparison as the crystal structure of its AlaRS has been obtained as it is not phylogenetically close to TuAlaRS as the authors show in Figure 5.
2. The introduction especially could use some editing:
Line 24, domains is used twice in the same sentence but in two different meanings.
Line 16: "except for a few exceptions" could be rephrased
3. Figure 3 is currently hard to read - can it be converted into either a color figure or can a legend be added?

Manuscript Number: COMMSBIO-22-3961

Article Title: A naturally occurring mini-alanyl-tRNA synthetase

Journal Name: *Communications Biology*

Dear Editor,

We would like to thank you and the reviewers for your careful consideration of the above-referenced manuscript. We have carefully revised the manuscript as suggested by the reviewers. Responses to the reviewer's comments are given below.

Reviewer's comments:

Reviewer 1:

Major

Q1: According to the complex structure of *A. fulgidus* tRNA:AlaRS, G3:U70 base pair and the hydrogen bonding contacts on the G3:U70 base pair are made with the major-groove N and minor-groove D residues. The authors should mention them more specifically in Introduction or Discussion.

A1: We have revised the introduction part. (page 3, lines 6-11)

Q2: We cannot tell that the loops (especially anticodon-loop) of Tupanvirus tRNA^{Ala} are smaller than normal (Page 4, Line 20-21 and also Fig. S1). Anticodon loop is normally composed of 7 nucleotides and "U-A" in Tupanvirus tRNA^{Ala} may be "U and A" (without base pair) as same as EctRNA^{Ala}. (I think "U and A" (without base pair) are more probable because Tupanvirus tRNA^{Ala} can be used in the ribosome.)

A2: We have revised Fig. S1A and the text. (page 4, lines 24-27)

Q3: "In contrast to EcAlaRS, which charged Ec-micro^{Ala} poorly (Fig. 2D), TuAlaRS charged this microhelix efficiently (Fig. 2C)" (Page 5, Line 6-7) is not appropriate. It depends on the concentration of the enzyme and the substrate used in the experiment. The author should modify the description. Because the authors also determined the kinetic parameters, the explanations should be dependent on the kinetics.

A3: We have revised the description. In these assays, we used 100 nM of AlaRS and 5 μM of microhelix^{Ala}. (page 5, lines 13-16)

Q4: The fact that deletion and substitution of the 3'-CCA end showed a comparable effect on TuAlaRS's binding gave the most impact on me in their paper. The authors should make more discussion on the recognition of the 3'-CCA end by this mini-AlaRS in terms of structure. The authors can at least compare the sequence-alignments of mini-AlaRS and *A. fulgidus* AlaRS (as shown in Fig. 1B). In addition, the specific residues responsible for the CCA recognition can be obtained from the complex structure of *A. fulgidus* tRNA:AlaRS is available. Therefore, the authors should use of them for discussion.

A4: According to the *A. fulgidus* AlaRS:tRNA^{Ala} complex, the amino acid residues ¹⁹³GGG¹⁹⁵ act as the route separator between reactive and non-reactive CCA-ends (Naganuma *et al.*, Nature 2014). Mutation of any of these three Gly residues causes mis-aminoacylation of G3:C70-containing tRNA^{Ala} (Miller *et al.*, Biochemistry 1991). Interestingly, the second and third Gly residues are conserved in TuAlaRS and other AlaRSs, implying that the remaining Gly residues play a similar role as the route separator. Moreover, unlike *E. coli* AlaRS, which binds the CCA-containing and CCA-deletion (or substitution) tRNAs^{Ala} with a similar affinity, TuAlaRS distinctly prefers the CCA-containing tRNA^{Ala} (Fig. 3AB), lending further support to the hypothesis that TuAlaRS has gained additional specific contacts with the acceptor stem through the CCA-end. (page 11, lines 7-16)

Q5: *Drosophila melanogaster* mitochondrial (Dm mt) AlaRS recognizes G2:U71. In this Dm mtAlaRS, an insertion of 27 amino acids in the region of the protein that contacts the acceptor stem played an important role of G2:U71 recognition (Lovato *et al.*, Mol Cell. 2004 843-851). Can the authors make a discussion about the similar insertion-based interpretation for their result, especially regarding the G1:U72 anti-determinant?

A5: In the case of tyrosyl-tRNA synthetase, an insertion peptide that splits the active site-containing domain determines the species-specific acylation. Substrate specificity can be switched by transplanting part of the insertion from human TyrRS into *E. coli* TyrRS and, reciprocally, by transplanting the counterpart from the *E. coli* enzyme into the human enzyme (Wakasugi, *et al.*, EMBO J 1998). Unfortunately, there is no insertion peptide found in the active site-containing domain of AlaRS. Instead, an anti-determinant, G1:U72, is found in the acceptor stem of the *C. elegans* mitochondrial tRNA^{Ala}. *E. coli* and Tupanvirus AlaRSs fail to charge this tRNA because of the presence of G1:U72. Charging is effectively blocked by the 4-ketooxygen of U72 in the major groove. However, adaptations make *C. elegans* mitochondrial AlaRS insensitive to the presence of the 4-ketooxygen. Extensive alanine-scanning mutagenesis of motif 2 of EcAlaRS provided little evidence for important contacts between tRNA^{Ala} and this segment of the protein (Davis *et al.*, Biochemistry 1994), which is largely consistent with the observation made in the cocrystal structure of *A. fulgidus* AlaRS:tRNA^{Ala} (Naganuma *et al.*, Nature 2014). Perhaps, it is the flexibility of the motif 2 loop, instead of the amino acid residues therein, that enables the *C. elegans* mitochondrial enzyme to accommodate the 4-ketooxygen. (page 10, lines 3-18)

Q6: Regarding the G:U recognition, the conserved N and D architecture of AlaRSs can select G:U in a straightforward (bacteria) or two different unconventional (eukarya/archaea) ways (reference #13). The authors should discuss (or at least speculate) on the role of P and T in TuAlaRS, which are located at the corresponding positions as N

and D, in considering above mentioned different modes for recognition of G:U pair in tRNA^{Ala}s from different species.

A6: The conserved N and D architecture of AlaRSs can select G:U in a straightforward (bacteria) or two different unconventional (eukarya/archaea) ways. In these scenarios, N and D either positively interact with G3:U70 or negatively select against non-G3:U70. However, mutation of P or T in TuAlaRS had little effect on G3:U70 selection, implying that they are not involved in G3:U70 recognition in any way. On the other hand, these two amino acids might play a role in the enzyme's structural stability or flexibility. To facilitate the acceptor stem binding, the enzyme might select the universal identity element through a new site/mode. (**page 9, lines 27-29**)

Minor

Q1: "GU base pair" should be written as "G:U base pair" (Page 3, Line 7; Page 6, Line 16; Page 6, Line 19; Page 9, Line 15).

A1: We have revised the text. (**page 3, line 8; page 6, lines 28 and 31; page 10, line 2**)

Reviewer 2:

Major

Q1: While the authors discuss the loss of C-Ala and provide experimental evidence for how TuAlaRS evolved around this loss, the lack of the editing domain is less explored. Loss of AlaRS editing in yeast causes misincorporation of serine - is increased sensitivity towards serine toxicity observed in the TuAlaRS-complemented yeast strains? Alternatively, can increased mischarging be observed in vitro? Could host AlaRS compensate for the lack of editing by TuAlaRS or might there even be a speculated evolutionary advantage of this lack in editing?

A1: To gain insight, we carried out complementation assays on 5-FOA with additional serine. The results showed that addition of extra serine (up to 5 mM) does not cause any toxic phenotype in the yeast knockout strain carrying TuAlaRS, suggesting that TuAlaRS has tolerable or low mischarging activity. Therefore, it is unnecessary for the host AlaRS to compensate for the lack of editing by TuAlaRS. (**Fig. S7 and page 8, lines 18-23**)

Q2: Does TuAlaRS charge the tRNA^{Ala} isoacceptors of its host? If not, this would help support the hypothesis that the complete set of aaRSs was acquired by Tupanviruses to enable orthogonality between virus and host translation. If experimental data is difficult to obtain, a comparison between all tRNAs could be added to Figure S1.

A2: We have carried out aminoacylation assays for TuAlaRS towards its host (*A. castellanii*) tRNA_n^{Ala}. The results showed that TuAlaRS can efficiently charge *A. castellanii* tRNA_n^{Ala}. (**Fig. S3 and page 5, lines 23-26**)

Minor:

Q1: *A. fulgidus* is mentioned without context - I am assuming it is used for comparison as the crystal structure of its AlaRS has been obtained as it is not phylogenetically close to TuAlaRS as the authors show in Figure 5.

A1: The phylogenetic tree shown in Fig. 5 demonstrated that TuAlaRS is actually originated from a bacterial origin. *A. fulgidus* AlaRS structure was used herein for comparison because its crystal structure is available.

Q2: The introduction especially could use some editing:

Line 24, domains is used twice in the same sentence but in two different meanings.

Line 16: “except for a few exceptions” could be rephrased

A2: We have revised the text. (**page 2, lines 16 and 24-25**)

Q3: Figure 3 is currently hard to read - can it be converted into either a color figure or can a legend be added?

A3: We have added symbols into Fig. 3.

REVIEWERS' COMMENTS:

Reviewer #1 (Remarks to the Author):

The authors have satisfactorily responded to all my questions and made the necessary changes to the manuscript.

Reviewer #2 (Remarks to the Author):

The authors addressed all points. Figure 3 could still be more clear and I would advise that *A. fulgidus* would be formally introduced in the text but I have no objections to the manuscript's publication.

Manuscript Number: COMMSBIO-22-3961A

Article Title: A naturally occurring mini-alanyl-tRNA synthetase

Journal Name: *Communications Biology*

Dear Editor,

We would like to thank you and the reviewers for your careful consideration of the above-referenced manuscript. We have carefully revised the manuscript as suggested by the reviewers. Responses to the reviewer's comments are given below.

Reviewer's comments:

Reviewer 2:

Minor:

Q1: Figure 3 could still be more clear and I would advise that *A. fulgidus* would be formally introduced in the text but I have no objections to the manuscript's publication.

A1: We have revised Fig.3 and introduced *A. fulgidus* AlaRS in the introduction part. **(page 2, lines 24-28)**